# 5-Hydroxymaltol Derived from Beetroot Juice through *Lactobacillus* Fermentation Suppresses Inflammatory Effect and Oxidant Stress via Regulating NF-kB, MAPKs Pathway and NRF2/HO-1 Expression

**DOI:** 10.3390/antiox10081324

**Published:** 2021-08-23

**Authors:** Su-Lim Kim, Hack Sun Choi, Yu-Chan Ko, Bong-Sik Yun, Dong-Sun Lee

**Affiliations:** 1Interdisciplinary Graduate Program in Advanced Convergence Technology & Science, Jeju National University, Jeju 63243, Korea; ksl1101@jejunu.ac.kr (S.-L.K.); choix074@jejunu.ac.kr (H.S.C.); uchan@jejunu.ac.kr (Y.-C.K.); 2Subtropical/Tropical Organism Gene Bank, Jeju National University, Jeju 63243, Korea; 3Bio-Health Materials Core-Facility Center, Jeju National University, Jeju 63243, Korea; 4Practical Translational Research Center, Jeju National University, Jeju 63243, Korea; 5Faculty of Division of Biotechnology, College of Environmental and Bioresource Sciences, Jeonbuk National University, Gobong-ro 79, Iksan 54596, Korea; bsyun@jbnu.ac.kr; 6Faculty of Biotechnology, College of Applied Life Sciences, Jeju National University, SARI, Jeju 63243, Korea

**Keywords:** 5-hydroxymaltol, *Lactobacillus* fermentation, inflammation, lipopolysaccharide, cytokines, Nrf-2

## Abstract

Inflammation is the first response of the immune system against bacterial pathogens. This study isolated and examined an antioxidant derived from *Lactobacillus* fermentation products using cultured media with 1% beet powder. The antioxidant activity of the beet culture media was significantly high. Antioxidant activity-guided purification and repeated sample isolation yielded an isolated compound, which was identified as 5-hydoxymaltol using nuclear magnetic resonance spectrometry. We examined the mechanism of its protective effect on lipopolysaccharide (LPS)-induced inflammation of macrophages. 5-Hydroxymaltol suppressed nitric oxide (NO) production in LPS-stimulated RAW 264.7 cells. It also suppressed tumor necrosis factor α (TNF-α), interleukin (IL)-1β, and inducible nitric oxide synthase (iNOS) in the messenger RNA and protein levels in LPS-treated RAW 264.7 cells. Moreover, it suppressed LPS-induced nuclear translocation of NF-κB (p65) and mitogen-activated protein kinase activation. Furthermore, 5-hydroxymaltol reduced LPS-induced reactive oxygen species (ROS) production as well as increased nuclear factor erythroid 2–related factor 2 and heme oxygenase 1 expression. Overall, this study found that 5-hydroxymaltol has anti-inflammatory activities in LPS-stimulated RAW 264.7 macrophage cells based on its inhibition of pro-inflammatory cytokine production depending on the nuclear factor κB signaling pathway, inhibition of LPS-induced reactive oxygen species production, inhibition of LPS-induced mitogen-activated protein kinase induction, and induction of the nuclear factor erythroid 2–related factor 2/heme oxygenase 1 signaling pathway. Our data showed that 5-hydroxymaltol may be an effective compound for treating inflammation-mediated diseases.

## 1. Introduction

Innate, or nonspecific immune response is the first barrier against invading detrimental particles and organisms, and it involves macrophage and inflammatory biomolecules [1]. Inflammation occurs in all human tissue and has a protective effect against extrinsic pathogens and intrinsic injuries [2]. It is a good reaction in humans; however, excessive inflammation is associated with chronic disease: rheumatoid arthritis, atherosclerosis, and diabetes [3]. Macrophages occur in the whole-body tissues, where they ingest foreign materials and dead cells as well as respond to various infection signals [4]. They have toll-like receptors that serve as a first-line host defense against infection [4,5]. Lipopolysaccharides (LPSs) are major components of the cell wall of gram-negative bacteria and induce strong cellular signals of macrophages in the innate immune system [5,6]. They activate macrophages by binding toll-like receptor 4 and stimulate several intracellular signaling pathways, including the nuclear factor κB (NF-κB) pathway and three mitogen-activated protein kinase (MAPK) pathways, namely, extracellular signal-regulated kinases (ERK), c-Jun N-terminal kinases (JNKs)/stress-activated protein kinases, and p38 MAPK [7,8]. Macrophages are activated under LPS treatment and lead to excessive production of inflammatory mediators, cytokines and reactive oxygen species (ROS) [9,10,11]. Nitric oxide (NO) induced by NO synthase is a pro-inflammatory mediator and induces oxidative stress and inflammation [12]. The major pro-inflammatory mediators, namely, tumor necrosis factor-α (TNF-α) and interleukin-1β (IL-1β), are overexpressed on LPS-treated macrophages and promote pathogenesis of inflammatory diseases [12]. Heme oxygenase 1 (HO-1) plays a crucial role in anti-inflammation and iron homeostasis. Its expression has been regarded as an adaptive cellular response against inflammation and oxidative injury [13]. It is mediated by nuclear factor erythroid 2–related factor 2 (Nrf2), which has an important role in cellular protection against inflammation and oxidative stress.

Red beetroot is a vegetable whose root can be eaten and used as raw material in the food industry. It contains bioactive compounds including anthocyanin, betacyanin, flavonoids, and other active components. The most important bioactive phytochemical of red beet is betalain, a tyrosine-derived pigment. Betanin, a type of betalain, has been shown to be an effective radical scavenging antioxidant [14,15] and anti-inflammatory compound [16]. Our group previously isolated the mammosphere formation inhibitor beta-vulgarin from beet [17]. Beta-vulgarin also functions against fungi [18]. Red beet is used for the production of probiotic beet juice via lactic acid bacterial fermentation. The antioxidant properties of blackberry are known to be enhanced by the fermentation of lactic acid bacteria as well [19]. Building on these previous studies, we attempted to characterize beetroot as a source of lactic acid bacterial fermentation derivative containing enhanced antioxidant effects. Another study isolated 5-hydroxymaltol from the Turkish apple and determined its antioxidant activity [20]. By contrast, data on the anti-inflammatory effects of 5-hydroxymaltol in murine macrophages have not been reported to date.

In this study, we investigated the anti-inflammatory activity of a beet fermentation derivative using RAW 264.7 cells. We purified the component using anti-inflammatory activity–based purification and found that the purified compound, 5-hydroxymaltol, has an anti-inflammatory effect. To our knowledge, this is the first study to have shown that 5-hydroxymaltol has an anti-inflammatory effect on RAW 264.7 cells by regulating NF-κB, MAPK pathways, and Nrf2/HO-1 expression.

## 2. Materials and Methods

### 2.1. Chemicals

Silica gel (60, 0.035–0.2 mm) resin (MERCK, Darmstadt, Germany), Sephadex G-10 (40–120 μm) powder (Cytiva, Marlborough, MA, USA), and thin-layer chromatography (TLC) Silica gel 60 RP-18 F_254_ plate (Sigma, Burlington, IA, USA) were used for chromatography. TLC Silica gel 60 RP-18 was performed with water and methanol (1:1), and spots were detected by a UV detector (Sigma). LPS (lipopolysaccharide from *E. coli* O111:B4) was purchased from InvivoGen (San Diego, CA, USA). Chemicals, such as 5-hydroxymaltol, were obtained from Enamine (Kyiv, Ukraine).

### 2.2. Plant Materials

Beet was obtained from farmers (Jejusi, Jeju, Korea). The beets were cleaned with distilled water and ground with a blender (Shinil, Seoul, Korea). The beet sample (No. 2018_010) was stored at Jeju National University Bio-Health Materials Core-Facility Center (JeJu-Si, Korea).

### 2.3. Fermentation of Beet Using Lactic Acid Bacteria

*Lactobacillus rhamnosus* GG (KCTC 3237) was purchased from the Korean Collection for Type Cultures (KCTC, Seoul, Korea). Stock cultures were placed at −80 °C in de Man, Rogosa, and Sharpe (MRS) medium (Difco, Detroit, MI, USA) with 20% (*v*/*v*) glycerol. Bacterial fermentation was started in 1-L flasks containing 100 mL of the MRS medium and incubated at 37 °C at 180 rpm for 1 day. MRS media containing 1% beet powder were adjusted to pH 6.5. The precultured *L. rhamnosus* samples were inoculated to MRS media containing 1% beet powder and incubated at 37 °C at 180 rpm for 0 or 5 days.

### 2.4. Extraction and Purification

The beet powder (1 kg) was incubated with MRS media (50 L) containing *L. rhamnosus*, and the fermented media were centrifuged. Next, 50 L of cultured media was extracted with 50 L of ethyl acetate (EA) (Appendix A). The EA extract parts were concentrated and solubilized with methanol. The antioxidant bioassay-based isolation protocol is summarized in Figure 1B. It was performed using diphenyl-picrylhydrazyl (DPPH) scavenging activity assay. The methanol extracts were prepared in an RP-18 manual column (10 cm × 14 cm), and RP-18 resin-containing sample was fractionated with 0% (water), 10%, and 20% methanol (Appendix A). Three fractionated parts were isolated and examined for antioxidants. The 0% methanol-eluted part showed an antioxidant effect. The concentrated part was applied to a silica gel 60 column (30 mm × 450 mm) and fractionated with chloroform/methanol (10:1, *v*/*v*) (Appendix A). Each eluted fraction was examined by TLC; four fractions were isolated and examined using an antioxidant assay. The second fraction showed an antioxidant effect. Therefore, the second part was loaded to a Sephadex G-10^TM^ gel column (3 cm × 45 cm) and eluted with water (Appendix A). Each eluate was checked by TLC; two fractions were isolated and examined by an antioxidant assay. The second fraction showed an antioxidant effect. It was then charged to preparative C-18 TLC (glass plate; 10 cm × 20 cm) and developed in methanol/water solvent (1:1, *v*/*v*) in a TLC chamber. The plates were taken out after development, and the fraction was analyzed for fluorescence under a UV detector (UV_254_). Isolated bands were purified by scraping them off the glass plate using a knife and collected in a 50-mL conical tube. Subsequently, 20 mL of methanol was added to the isolated gel, and the mixture was vortexed and centrifuged. Each part was obtained and examined using an antioxidant assay (Appendix A). The active part was charged to preparatory high-performance liquid chromatography (HPLC). HPLC analysis was performed using Shimadzu HPLC (Shimadzu, Tokyo, Japan). HPLC separation was performed using an RP-18 (10 mm × 250 mm) column. The isolated sample was sieved through a 0.2 µm filter. The injection volume was 0.5 mL, the flow rate was 2.5 mL/min and the column temperature was room temperature at 220 and 254 nm. The mobile phase consisted of water (solvent A) and acetonitrile (solvent B). For gradient elution, solvent B was initially set at 0%, increased to 5% at 20 min, and increased to 100% at 30 min. The isolated peak was shown at 36 min (Figure 1C and Appendix A).

### 2.5. Structure Analysis of the Isolated Sample

The molecular structure of the purified compound was analyzed by electrospray ionization mass spectrometry as well as one- and two-dimensional nuclear magnetic resonance (NMR) data. Mass measurement determined the molecular weight to be 142 Da, which indicated a quasi-molecular ion peak at *m*/*z* 143.1 [M+H]^+^ (Appendix A). The ^1^H-NMR data measured in deuterated methanol showed signals at an aromatic methine at δ 7.84 and a methyl at δ 2.30. In the ^13^C-NMR data, six carbon peaks including a carbonyl carbon at δ 170.3, three oxygenated sp^2^ carbons at δ 151.8, 145.8, and 142.9, an oxygenated sp^2^ methine carbon at δ 140.4, and a methyl carbon at δ 14.5 were evident (Appendix A). Two proton-bearing carbons were assigned by the heteronuclear multiple quantum coherence (HMQC) spectrum (Appendix A). Further structural elucidation was performed using the heteronuclear multiple bond coherence spectrum (HMBC) spectrum, which showed long-range correlations from the methine proton at δ 7.84 to the carbons at δ 170.3, 151.8, and 145.8 as well as from the methyl protons at δ 2.30 to the carbons at δ 151.8 and 142.9 (see Appendix A). Therefore, the molecular structure of the isolated compound was identified as 5-hydroxymaltol (Figure 2).

### 2.6. Antioxidant Assay

Free radical scavenging activity was examined via DPPH assay [21]. The increasing concentrations of 5-hydroxymaltol (0, 100, 200, 400, 500 µM) and 10 mM N-acetyl-L-cysteine (NAC) were mixed with 200 µM DPPH solution (4 mM stock, methanol) in a 96-well plate and incubated at room temperature for 30 min. The absorbance at OD_517_ was measured using a VersaMax ELISA (enzyme-linked immunosorbent assay) reader (Molecular Devices, San Jose, CA, USA). The DPPH scavenging activity was calculated to the follows:(1)$$\mathrm{DPPH}\;\mathrm{scavenging}\;\mathrm{activity}\;(\%)=\left[1-\left(\frac{\mathrm{Sample}-\mathrm{Blnak}}{\mathrm{Control}}\right)\right]\times 100$$
where sample is the absorbance of the sample, Control is the absorbance of the control with DPPH, and Blank is the absorbance of the sample without DPPH.

### 2.7. Cell Line and Culture Conditions

RAW 264.7 macrophage cells were obtained from the Korea Cell Line Bank (Seoul, Korea). The macrophage cells were incubated in high-glucose Dulbecco’s modified Eagle’s medium (DMEM) supplemented with 10% fetal bovine serum (Hyclone, Logan, UT, USA), and 1% penicillin/streptomycin (Hyclone). The cells were incubated at 37 °C in an atmosphere of 5% CO_2_.

### 2.8. Cell Proliferation

RAW 264.7 cells (3 × 10^5^ cells/mL) were seeded in a 96-well plate and cultured for 1 day. The cells were treated with increasing concentrations of 5-hydroxymaltol (0, 50, 100, 200, 400, 500, 750 and 1000 µM) for 1 day. Cell viability was examined using CellTiter 96 AQueous One Solution (Promega, Madison, WI, USA) following the manufacturer’s protocol. After the culture medium and aqueous solution were mixed (5:1), 100 μL of mixture was added to a 96-well plate and reacted at 37 °C for 2 h. The absorbance was estimated using a VersaMax ELISA reader at OD_490_.

### 2.9. NO Assay

RAW 264.7 cells (3 × 10^5^ cells/mL) were cultured in a 6-well plate and incubated for 1 day. The cells were treated with different concentrations of 5-hydroxymaltol (0 and 500 µM) without LPS or with increasing concentrations of 5-hydroxymaltol (0, 200, 500 and 1000 µM) with LPS (1 µg/mL) for 24 h. The measurement of NO was assessed in supernatant culture medium using NO Plus Detection kit (iNtRON Biotechnology, Gyeonggi, Korea). One hundred microliters of the culture medium or nitrite standard was induced pre-reaction by adding 50 µL of N1 buffer (sulfanilamide in buffer) to each well of a 96-well plate. The 96-well plate was then incubated for 20 min, and the mixture was reacted with 50 µL of N2 buffer (naphthyl-ethylenediamine in buffer). After the mixture was incubated for 10 min, NO formation was measured with the absorbance value at OD_560_ nm using a VersaMax ELISA reader.

### 2.10. RT-qPCR

RAW 264.7 cells were cultured in a 6-well plate with 5-hydroxymaltol (1000 µM) and treated with LPS (1 µg/mL) for 1 day. Total RNA was purified using TaKaRa MiniBEST RNA Extraction Kit (TaKaRa, Kyoto, Japan) according to the manufacturer’s protocol. Reverse transcription quantitative polymerase chain reaction (RT-qPCR) was performed using RNA-direct™ SYBR^®^ Green Realtime qPCR Master Mix (TOYOBO, Osaka, Japan). RT-qPCR mixture contained 10 μL of RNA-direct SYBR Green Realtime qPCR Master Mix, 1 μL of 50 mM Mn(OAc)_2_, 2 μL of template RNA (100 ng/μL), 2 μL of specific primer-F (10 ng/μL), 2 μL of specific primer-R (10 ng/μL), and 3 μL of nuclease-free H_2_O. The primers that were used to perform RT-PCR were purchased from Bioneer (Daejeon, Korea).

### 2.11. ELISA

RAW 264.7 macrophage cells were cultured in a 6-well plate. The attached cells were pretreated with increasing concentrations of 5-hydroxymaltol (0, 200, 500 and 1000 µM) for 24 h and stimulated with LPS (1 µg/mL) for 24 h. The measurement of TNF-α and IL-1β was assessed in supernatant medium. Based on the manufacturers’ protocols, the amount of IL-1 β was examined using IL-1β Mouse ELISA Kit (Invitrogen, Carlsbad, CA, USA), whereas the amount of TNF-α was measured using ELISA MAX™ kit (BioLegend, San Diego, CA, USA).

### 2.12. Immuno Blot Analysis

RAW 264.7 macrophage was pretreated with 5-hydroxymaltol for 1 h and stimulated with LPS (1 µg/mL) for 30 min. The macrophage cells were harvested and lysed using a radioimmunoprecipitation assay buffer (Thermo Fisher Scientific, Waltham, MA, USA). Each sample was separated by sodium dodecyl sulfate–polyacrylamide gel electrophoresis and electro-transferred to Immobilin-FL polyvinylidene fluoride (PVDF) membranes (Millipore, Burlington, MA, USA). After blocking with Odyssey^®^ Blocking Buffer (LI-COR, Lincoln, NE, USA) for 1 h, the blot was incubated overnight with the primary antibodies at 4 °C. After being washed, the blots were incubated with diluted IRDye 680- and IRDye 800-labeled secondary antibodies using Odyssey blocking buffer containing 0.2% Tween-20/0.01% sodium dodecyl sulfate for 60 min. The protein bands were detected with an Odyssey CLx imaging machine (LI-COR). The Primary antibodies against pp38, JNK, pJNK, ERK, pERK, p65, Nrf2 and HO-1 were obtained from Cell Signaling Technology (Beverly, MA, USA). The antibodies of p38 and β-actin were purchased from Santa Cruz Biotechnology, Inc (Dallas, TX, USA).

### 2.13. Measurement of ROS Activity Using CellROX Green Dye

The ROS were measured using CellROX Green probe (Life Technologies, Carlsbad, CA, USA) according to the manufacturer’s protocol. RAW 264.7 cells (2 × 10^6^ cells/10 mL) were cultured in a plate and incubated for 1 day. The macrophage cells were pretreated with 5-hydroxymaltol and NAC for 1 h and treated with LPS (1 µg/mL) for 30 min. Next, the cells were treated with CellROX green reagent. After the cells were incubated for 10 min at 37 °C, the medium was discarded and the cell were washed with 1× phosphate-buffered saline. The macrophage cells were measured using a fluorescence microscope (Lionheart, Biotek, VT, USA).

### 2.14. Statistical Analysis

Data were analyzed using one-way analysis of variance. All experimental values were expressed as mean ± standard deviation. *p* values were evaluated using GraphPad Prism 7 (GraphPad Software, San Diego, CA, USA), and *p* < 0.05 was considered statistically significant.

## 3. Results

### 3.1. Isolation of an Antioxidant Compound from Beet Fermentation Using L. rhamnosus GG

We cultured *L. rhamnosus* using MRS media containing 1% beet powder. After 5 days of fermentation, we examined the HPLC peaks using extracts of supernatant media with EA (cultured media: EA = 1:1, *v*/*v*). We found a new peak on day 5 culture (red box in Figure 1A). To find an antioxidant, we performed an antioxidant assay using the sample peak (red box). The new peak showed antioxidant effect. The antioxidant-guided purification protocol is summarized in the Figure 1B. The fermented beet broth was isolated using EA and cultured broth (1:1, *v*/*v*); subsequently the EA extracts were isolated using C-18 open column, silica gel chromatography, a Sephadex^TM^ G-10 gel chromatography, C-18 preparatory TLC, and preparatory HPLC (Figure 1B and Appendix A). The isolated compound is shown in Figure 1C.

### 3.2. Structural Analysis of Purified Compound and Effect of the Isolated Compound on Antioxidant Activity and Cell Proliferation

The isolated compound was identified as 5-hydroxymaltol using NMR and Mass spectrometry data (Figure 2A and Appendix A). First, we examined the antioxidant effect of 5-hydoxylmaltol in macrophage cell line, RAW 264.7 by DPPH scavenging activity assay. DPPH scavenging activity was increased to 42.5% at a high concentration of 500 μM. 5-hydroxymaltol showed strong antioxidant effect in concentration-dependent manner (Figure 2B). RAW 264.7 macrophage cells were incubated with several concentrations of 5-hydroxymaltol (up to 1000 μM). 1000 μM of 5-hydroxymaltol did not show any difference in cell proliferation (Figure 2C).

### 3.3. Effect of 5-Hydoxylmaltol on NO Production and Inducible NO Synthase Expression in RAW 264.7 Macrophage Cells

NO is a biological mediator and plays a role in LPS-stimulated macrophage. In this study, NO production was estimated at LPS-treated RAW 264.7 cells. To test the inhibitory effect of 5-hydoxylmaltol on LPS-induced NO production, we treated RAW 264.7 macrophage cells with 5-hydoxylmaltol (200, 500 and 1000 μM) for 24 h with or without LPS. The results showed that LPS-induced NO production and 5-hydoxylmaltol reduced LPS-induced NO production. LPS-stimulation induced a 51-fold increase in NO concentration, whereas 5-hydroxymaltol-treated cells showed corresponding reductions to 34%, 41%, and 54% compared with the LPS-treated control (Figure 3A). We determined the protein level of inducible NO synthase (iNOS) using Western blot. LPS-stimulated RAW 264.7 cells increased iNOS protein expression, whereas 5-hydroxylmaltol highly suppressed this. The transcript level of iNOS was also examined with LPS treatment in RAW 264.7 cells. LPS increased iNOS transcript levels, whereas 5-hydroxylmaltol reduced the increased levels of iNOS transcripts. LPS treatment increased the messenger RNA (mRNA) and protein levels of iNOS by 21- and 14-fold, respectively, whereas 5-hydroxymatol (1000 μM) downregulated level of mRNA and protein of iNOS by 46 and 61%, respectively, compared with the LPS-treated control (Figure 3B). Our data suggest that 5-hydroxymaltol inhibits NO production via the downregulation of iNOS.

### 3.4. 5-Hydroxymaltol Suppressed the LPS-Stimulated TNF-α and IL-1β Production and mRNA Level in RAW 264.7 Cells

LPS-activated macrophages produce pro-inflammatory cytokines and growth factor [6]. We tested the effect of 5-hydroxymaltol on elevated levels of TNF-α and IL-1β at the protein and transcript levels in RAW 264.7 cells using ELISA and RT-qPCR. LPS treatment increased mRNA and protein levels of IL-1β by 101- and 1.6-folds, respectively, whereas 5-hydroxymatol (1000 μM) downregulated them by 61% and 23%, respectively, compared with the LPS-treated control (Figure 4A). LPS treatment increased mRNA and protein levels of TNF-α by 5.9- and 11.9-fold, respectively, whereas 5-hydroxymatol (1000 μM) downregulated them by 42% and 43%, respectively, compared with the LPS-treated control (Figure 4B). Our data suggest that 5-hydroxymaltol inhibits IL-1β and TNF-α cytokine production via the downregulation of IL-1β and TNF-α gene (Figure 4B). Moreover, LPS treatment significantly increased the production of TNF-α and IL-1β in RAW 264.7 cells, whereas, 5-hydroxymaltol treatment reduced LPS-stimulated production of TNF-α and IL-1β in a concentration-dependent manner (Figure 4A,B). LPS treatment increased the transcripts levels of TNF-α and IL-1β genes, whereas 5-hydroxymaltol pre-treatment downregulated LPS-stimulated TNF-α and IL-1β gene expression (Figure 4A,B). Our data showed that 5-hydroxymaltol reduces immune reaction of RAW 264.7 cells by inhibiting LPS-stimulated cytokine production.

### 3.5. 5-Hydroxymaltol Suppressed LPS-Stimulated NF-κB Activation in RAW 264.7 Cells

The nuclear translocation of NF-κB increases the expression of inflammation-related genes in macrophage cells [22]. This study examined the ability of 5-hydroxymaltol to suppress the nuclear translocation of NF-κB. In LPS-treated cells, the nuclear translocation of NF-κB increased by 8.6-fold compared with the control, whereas 5-hydroxymatol downregulated the nuclear p65 protein by 73% compared with the LPS-treated control. We examined cytosolic inhibitor κB (IκB), which is a regulator of NF-κB. 5-Hydroxymaltol pretreatment blocked the downregulation of IκB-α in LPS-treated cells. The cytosolic IκB increased by 2-fold compared with the LPS-treated control. Our studies show that LPS treatment induces nuclear translocation of NF-κB (p65) with downregulation of IκB-α. Moreover, 5-hydroxymaltol pretreatment inhibited nuclear translocation of NF-κB (p65) and blocked the downregulation of IκB-α in LPS-treated cells (Figure 5). Our results show that 5-hydroxymaltol acts as a suppressive agent of LPS-induced NF-κB activation in RAW 264.7 macrophage cells.

### 3.6. 5-Hydroxymaltol Inhibited LPS-Stimulated MAPK Activation in RAW 264.7 Cells

MAPKs regulate cellular function, proliferation, gene expression, cell survival, and apoptosis [23]. In this study, the effect of 5-hydroxymaltol on LPS-stimulated phosphorylation of MAPKs were determined using immunoblot analysis. We observed that LPS treatment induced the phosphorylation levels of ERK, p38, and JNK to 4.8-, 3- and 1.9-folds, respectively compared with control, whereas 5-hydroxymatol downregulated them by 69%, 73%, and 59%, respectively, compared with the LPS-treated control (Figure 6). Our data show that 5-hydroxymaltol inhibited LPS-stimulated phosphorylation of ERK, p38, and JNK (Figure 6), and that 5-hydroxymaltol inhibited the inflammatory response of LPS-induced RAW 264.7 cells by inhibiting MAPK signals.

### 3.7. 5-Hydroxymaltol Decreased LPS-Induced ROS Production in RAW 264.7 Cells

LPS-stimulated macrophages induced ROS accumulation and oxidative stress [10,24]. In this study, the effect of 5-hydroxymaltol on accumulation of cellular ROS in LPS-induced RAW 264.7 cells was examined using CellROX Green dye. Figure 7 showed that LPS-treatment increased cellular density of CellROX Green and 5-hydroxymaltol and N-acetyl cysteine (NAC) reversed the LPS-induced increases in cellular ROS contents. Fluorescent intensity was increased by 13.6-fold in LPS-treated cells and decreased by 87% in 5-hydroxymaltol–treated cells compared with the LPS-treated control. Our data show that 5-hydroxymaltol has a strong ROS scavenging effect on RAW 264.7 cells.

### 3.8. 5-Hydroxymaltol Increased Protein Exprerssion of HO-1 and Nrf2 in RAW 264.7 Cells

The Nrf2 and HO-1 signaling pathway is an antioxidant system that reduces oxidative stress in animal models [25,26]. We examined whether antioxidant effect of 5-hydroxymaltol is involved in Nrf2 and HO-1 signaling pathway. 5-hydroxymaltol treatment increased protein expression of HO-1 and Nrf2 to 12 h by 3.9- and 3.8-fold, respectively. Our data showed that 5-hydroxymaltol treatment increased the protein levels of Nrf2 and HO-1 in a time-dependent manner (Figure 8A). We also analyzed the cytosolic and nuclear fractions for Nrf2 protein under 5-hydroxymaltol and found that the nuclear Nrf2 protein level significantly increased to 1.9-fold. The Western blot data of the cytosolic and nuclear fractions using RAW 264.7 cells showed that 5-hydroxymaltol treatment significantly increased the nuclear Nrf2 protein level (Figure 8B). Our data showed that antioxidant effect of 5-hydroxymaltol is associated with the Nrf2/HO-1 signaling pathway.

## 4. Discussion

The vegetable beetroot (*B. vulgaris*) is also known as red beet or, simply, beet. Its essential nutrients are fiber, vitamin B9, manganese, iron, and vitamin C. Beetroot and beet juice help improve blood flow, reduce blood pressure, and enhance exercise performance. Beetroot supplements have beneficial effects on inflammation, oxidative stress, and cognition function [27]. Beetroots have been described as a potential substrate for the production of probiotic juice using lactic acid bacteria [28]. Research has shown that beetroot and fermented beetroot juices have different chemical properties and different effects on cancer cells [29]. The antioxidant and anticoagulant activities of natural compounds can be increased by fermentation using lactic acid bacteria [30]. In this study, which is the first of its kind to be reported in the literature, we isolated an antioxidant through *Lactobacillus* fermentation using culture media with 1% beetroot powder. Activity-based purification and repeated sample isolation led to the isolation of a compound from lactic acid bacteria cultured media with beetroot powder. We showed that 5-hydroxymaltol from fermented broth has an anti-inflammatory effect on LPS-treated RAW 264.7 cells.

We investigated the protective effect of 5-hydroxymaltol in LPS-treated RAW 264.7 cells. LPS derived from gram-negative bacteria was used to make a model system to examine the ability of specific compounds to suppress the inflammatory response [31]. During an inflammatory response, LPS-activated macrophage produces inflammatory cytokines TNF-α and IL-1β. In this study, 5-hydroxymaltol reduced LPS-induced NO production, LPS-activated TNF-α, as well as IL-1β secretion and gene expression; it also inhibited LPS-induced inflammation.

The NF-κB is an important mediator of inflammatory responses and induces the expression of various pro-inflammatory genes, including cytokines and chemokines genes [32]. Compounds target the reduction of the nuclear translocation of NF-κB to inhibit the inflammatory response. Our results indicated that 5-hydroxymaltol not only inhibited LPS-stimulated nuclear translocation of NF-κB but also reduced gene expression in the inflammatory response. Studies have shown that NF-κB regulates the LPS-activated MAPK signaling pathway [22,33]. We thus examined whether 5-hydroxymaltol inhibits LPS-induced MAPK activation (ERK1/2, JNK, and p38). Our data showed that 5-hydroxymaltol suppressed the LPS-induced phosphorylation of three MAPKs, indicating that 5-hydroxymaltol inhibits LPS-induced inflammation by inhibiting MAPK activation and NF-κB translocation.

LPS stimulate ROS production in macrophages and induced the production of inflammatory mediators and cytokines [34]. ROS acts as a signal mediator in receptor-mediated signaling. The source of ROS in receptor-mediated signaling is lipoxygenase (LOXs) and nicotinamide adenine dinucleotide phosphate oxidases (NOXs). LPS stimulation has also been reported to reduce ROS production and NF-κB activation [35]. Our data indicated that 5-hydroxymaltol reduced LPS-induced ROS accumulation of macrophage and that blocking ROS production is an important factor in the anti-inflammatory effect of 5-hydroxymaltol. Nrf2 is known as the main regulator of cellular antioxidant defenses and as a suppressor of inflammation. The anti-inflammatory activity of Nrf2 signaling can be manifested by its upregulation of cellular antioxidant defenses via antioxidant response transcription. Nrf2 has been reported to suppress transcriptional upregulation of pro-inflammatory cytokines and inflammation [36]. Nrf2/HO-1 axis has been found to reduce oxidative stress in cellular and animal systems [37]. Nrf2/Maf complex proteins bind to the antioxidant response element to increase the transcription of target genes, including HO-1. In addition, bilirubin derived from HO-1 has antioxidant and cytoprotective effects. Our data suggest that 5-hydroxymaltol induced the protein levels of Nrf2, and HO- and elevated Nrf2/HO-1 signaling induce anti-inflammatory response of macrophage to LPS treatment by reducing oxidative stress. Our data also indicated that LPS treatment increased the transcript and protein levels of iNOS and IL-1β, whereas 5-hydroxymaltol reduced them. However, the effects of 200, 500, and 1000 μM of 5-hydroxymaltol on the LPS-induced transcript and protein levels of iNOS and IL-1β did not differ. We do not know the exact mechanism at play yet, but a high concentration of 5-hydroxymaltol may induce posttranscriptional and translational regulation of iNOS and IL-1β genes. Moreover, our experimental data showed that 500 μM (71 mg/kg) and 1000 μM (142 mg/kg) of 5-hydroxymaltol had an anti-inflammatory effect in RAW 264.7 cells. Other research groups have reported that 50 and 100 mg/kg of maltol inhibited oxidative stress, inflammation, and apoptosis in in vivo mouse experimental models [38]. We surmise that 500 μM (71 mg/kg) and 1000 μM (142 mg/kg) of 5-hydroxymaltol will have an anti-inflammatory effect on a mouse model system.

5-Hydroxymaltol, a derivative of maltol, is found in *Penicillium echinulatum*, toast oak, and honey [20,39,40]. It has previously been isolated from the Turkish apple and found to have antioxidant activity [20]. However, studies on the anti-inflammatory effects of 5-hydroxymaltol in murine macrophage cells have not been reported to date. The results of our study showed that 5-hydroxymaltol exhibits its anti-inflammatory effect on macrophages not only by inhibiting ROS production, MAPK activities, and NF-κB (p65) nuclear translocation but also by enhancing Nrf2/HO-1 signaling (Figure 9).

## 5. Conclusions

We isolated and examined antioxidant derived from *Lactobacillus* fermentation using cultured media with 1% beet powder. The anti-oxidant activity of cultured beet media was significantly high. Antioxidant activity-based purification and repeated sample preparation led to the isolation of a compound. Using NMR spectrometry, we identified the purified compound as 5-hydoxymaltol. We examined the mechanism of its protective effect on lipopolysaccharide-induced inflammation of macrophage. 5-Hydroxymaltol exhibits its anti-inflammatory activities in LPS-treated macrophages by inhibiting inflammatory cytokine production depending on the NF-κB signaling pathway, inhibiting LPS-induced ROS accumulation, suppressing LPS-stimulated MAPK activation, and activating the Nrf2/HO-1 signaling pathway. Our data indicate that 5-hydroxymaltol may be useful in treating inflammation-mediated diseases

## Figures and Tables

**Figure 1 antioxidants-10-01324-f001:**
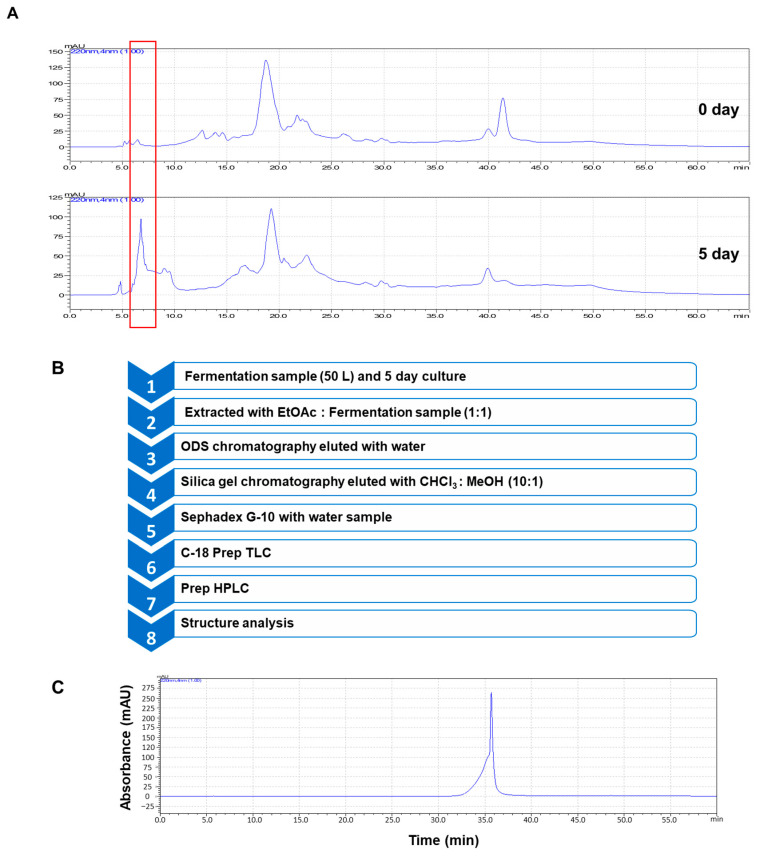
HPLC data of *Lactobacillus* cultured media with beet juice and purification of anti-inflammatory compound derived from beet juices by *Lactobacillus* fermentation. (**A**) HPLC analysis of *Lactobacillus* fermentation using beet (0- and 5-day cultures). Red box: new peak. (**B**) Isolation of the anti-inflammatory compound. (**C**) HPLC chromatogram of the isolated anti-inflammatory compound.

**Figure 2 antioxidants-10-01324-f002:**
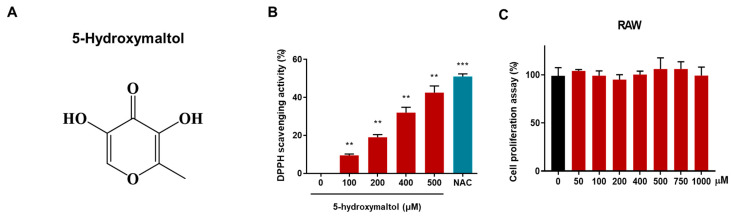
Chemical structure of anti-inflammatory compound derived from beet fermentation using *Lactobacillus rhamnosus,* as well as cell viability and free radical scavenging activity assays. (**A**) Molecular structure of 5-hydroxymaltol. (**B**) DPPH radical scavenging activity of 5-hydroxymaltol with methanol. (**C**) RAW 264.7 macrophage cells were exposed to several concentrations of 5-hydroxymatol for 1 day and cell proliferation was examined using MTS assay. These data are shown as mean ± SD, *n* = 3. ** *p* < 0.01; *** *p* < 0.001.

**Figure 3 antioxidants-10-01324-f003:**
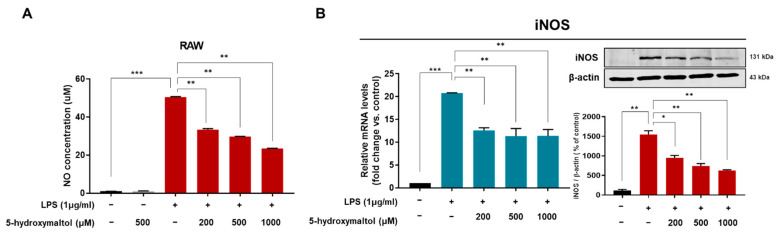
Effect of 5-hydroxymaltol on LPS-stimulated NO production and expression of iNOS in RAW 264.7 cells. (**A**) RAW 264.7 cells (6 × 10^5^ cells/well in 6-well plates) were left untreated pre-treated with 5-hydroxymaltol (200, 500 and 1000 μM) prior to treatment with LPS (1 μg/mL) for 1 day. (**B**) Total RNA was isolated from RAW 264.7 macrophage cells with or without the indicated concentration of 5-hydroxymaltol and then stimulated with LPS (1 μg/mL) for 1 day. The mRNA level of iNOS was examined by reverse transcription quantitative polymerase chain reaction as described in Section 2. Protein lysates were isolated from the cells with or without the indicated concentrations of 5-hydroxymaltol and then treated with LPS (1 μg/mL) for 24 h. The values are expressed as mean ± standard deviation, *n* = 3. * *p* < 0.05; ** *p* < 0.01; *** *p* < 0.001.

**Figure 4 antioxidants-10-01324-f004:**
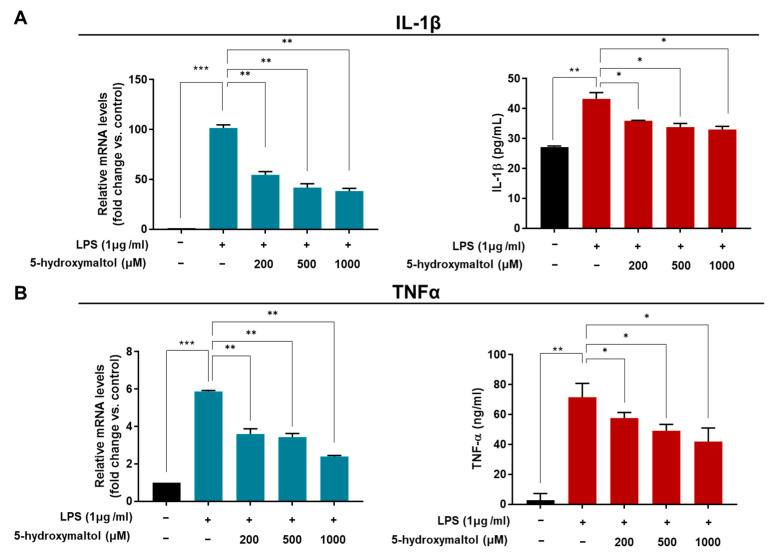
Effect of 5-hydroxymaltol on LPS-induced IL-1β and TNF-α production as well as mRNA expression of IL-1β and TNF-α in RAW 264.7 cells. (**A**) RAW 264.7 cells were left untreated or pretreated with 5-hydroxymaltol (200, 500 and 1000 μM) prior to stimulation with LPS (1 μg/mL) for 1 day. Total RNA was isolated from RAW 264.7 cells. The mRNA level of IL-1β was determined using reverse transcription quantitative polymerase chain reaction. IL-1β production in culture media was quantified using an enzyme-linked immunosorbent assay kit. (**B**) RAW 264.7 cells were left untreated or pretreated with 5-hydroxymaltol (200, 500 and 1000 μM) to stimulation with LPS (1 μg/mL) for 1 day. Total RNA was isolated from RAW 264.7 cells. The mRNA level of TNF-α was determined using reverse transcription quantitative polymerase chain reaction. TNF-α production in culture media was quantified using an enzyme-linked immunosorbent assay kit. The values are expressed as mean ± standard deviation, *n* = 3. * *p* < 0.05; ** *p* < 0.01; *** *p* < 0.001.

**Figure 5 antioxidants-10-01324-f005:**
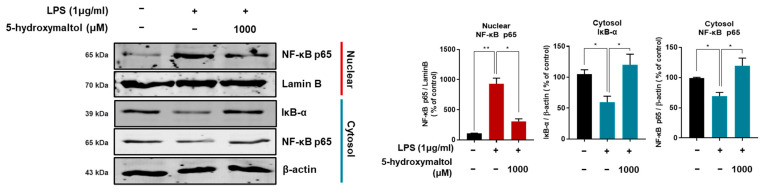
5-hydroxymaltol inhibits LPS-induced nuclear translocation of NF-κB (p65) in RAW 264.7 cells. Macrophage cells were treated with 1000 μM of 5-hydroxymaltol for 1 h prior to treatment with LPS (1 μg/mL) for 30 min. Nuclear and cytosolic fractions were analyzed using sodium dodecyl sulfate–polyacrylamide gel electrophoresis, followed by immunoblotting using the indicated antibodies. Lamin B and β-actin served as the internal controls for nuclear and cytosolic fractions. The values are expressed as mean ± standard deviation, *n* = 3. * *p* < 0.05; ** *p* < 0.01.

**Figure 6 antioxidants-10-01324-f006:**
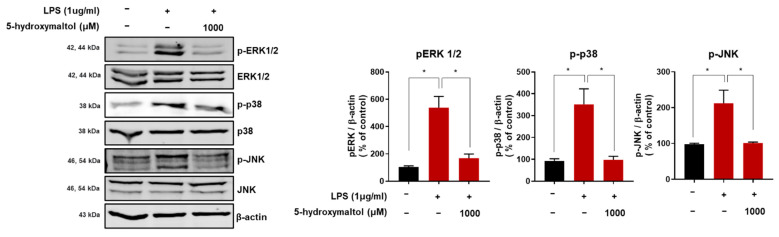
Effect of 5-hydroxymaltol on LPS-stimulated MAPKs activation in RAW 264.7 cells. The RAW 264.7 cells were cultured with 1000 μM of 5-hydroxymaltol for 1 h prior to treatment with LPS (1 μg/mL) for 30 min. Total protein extracts were analyzed using sodium dodecyl sulfate–polyacrylamide gel electrophoresis, followed by immunoblotting using several antibodies. β-actin served as the internal control for total proteins. MAPK, ERK, p38, and JNK signals were detected. The values are expressed as mean ± standard deviation, *n* = 3. * *p* < 0.05.

**Figure 7 antioxidants-10-01324-f007:**
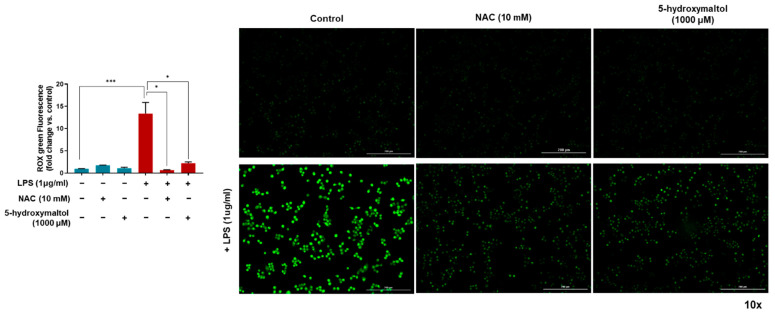
Effect of 5-hydroxymaltol on LPS-stimulated reactive oxygen species accumulation on RAW 264.7 macrophage cells. The effect of 5-hydroxymaltol on reactive oxygen species accumulation was determined using CellROX Green dye assays. Macrophage cells were treated with 1000 μM of 5-hydroxymaltol and NAC (10 mM) for 1 h prior to treatment with LPS (1 μg/mL) for 30 min. Images were captured using microscopy at 10× magnification and are representative photos (scale bar = 100 μm). The values are expressed as mean ± standard deviation, *n* = 3. * *p* < 0.05; *** *p* < 0.001.

**Figure 8 antioxidants-10-01324-f008:**
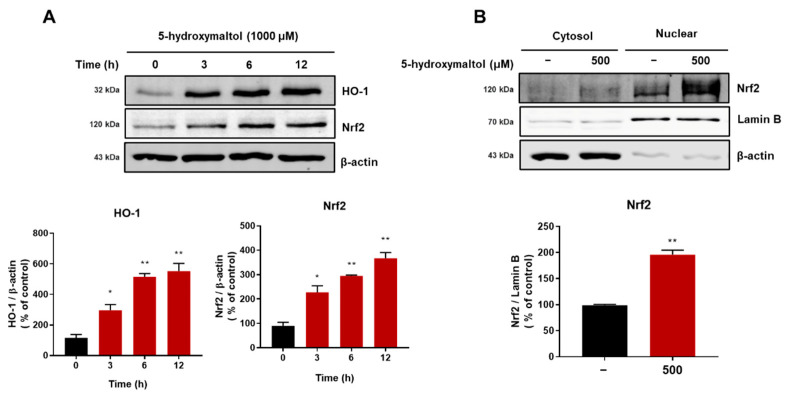
Increase of Nrf2 and HO-1 protein levels by 5-hydroxymaltol in RAW 264.7 cells. Macrophage cells were cultured with 1000 μM of 5-hydroxymaltol. (**A**) Total proteins were separated by 10% sodium dodecyl sulfate–polyacrylamide gel electrophoresis and then analyzed via Western blot using Nrf2 and HO-1 antibodies. β-Actin served as the internal control. (**B**) Macrophage cells were treated with 500 μM of 5-hydroxymaltol for 12 h. Nuclear and cytosolic fractions were analyzed using sodium dodecyl sulfate–polyacrylamide gel electrophoresis, followed by immunoblotting using several antibodies. Lamin B and β-actin served as the internal controls for the nuclear and cytosolic fractions. The values are expressed as mean ± standard deviation, *n* = 3. * *p* < 0.05; ** *p* < 0.01.

**Figure 9 antioxidants-10-01324-f009:**
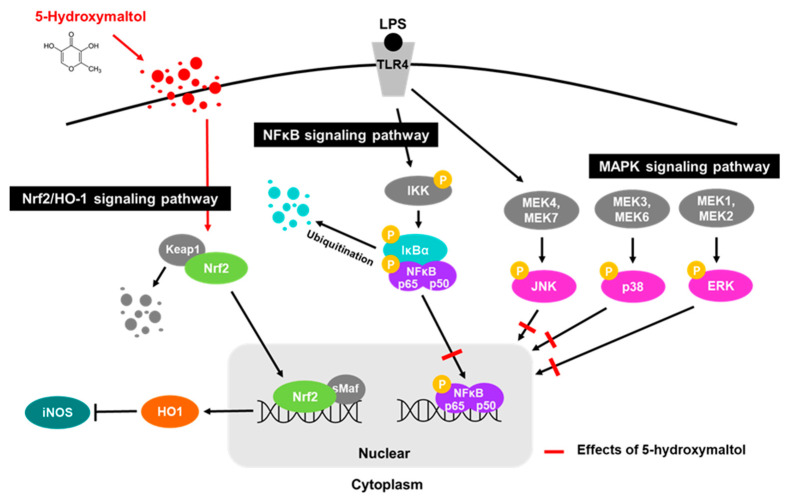
Scheme of the mechanisms of the protective activity of 5-hydroxymaltol on LPS-stimulated inflammation.

## Data Availability

The data presented in this study are available on request from the corresponding author. The raw data supporting the conclusions of this manuscript will be made available by the authors to any qualified researcher.

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
