# Peer review of "5-Hydroxymaltol Derived from Beetroot Juice through Lactobacillus Fermentation Suppresses Inflammatory Effect and Oxidant Stress via Regulating NF-kB, MAPKs Pathway and NRF2/HO-1 Expression"

_antioxidants, 2021, doi:10.3390/antiox10081324_

Round 1

Reviewer 1 Report

General comments:

The authors isolated a novel compound, so-called 5-hydroxymaltol, produced by fermentation from beet root juice by Lactobacillus rhamnosus, and characterized it as an anti-oxidative agent. In addition to the interesting data presented in the manuscript I recommend the following changes:

Major comments:

  • The fermentates of beet root were separated by its anti-oxidative capacities, as described in page 3, line 117. Please specify how the anti-oxidative effect has been analysed?
  • Please improve the quality of figure 1 and increase font size.
  • Please show staining controls for immunofluorescence.
  • Please show uncropped Western Blots.
  • Figure 3: For mRNA expression typically Quantitative analysis is performed using the 2-ΔΔCT method and the results are expressed as fold changes. Which analyses has been used in the presented study?
  • Can the authors explain the mechanism of 5-hydrxymaltol? How does it pass the cell membrane? Does 5-hydrxymaltol bind to a membrane receptor?

Minor comments:

  • Page 1, line 21: Please change “power” to “powder”.
  • Page 2, line 44: Please change “abnormal inflammation” to “excessive inflammation”.
  • Page 2, line 59: “The major pro-inflammatory mediators, […] are overexpressed on LPS-treated macrophage and give pathogenesis of inflammatory diseases”. Please change “give” to “promote or drive etc.”
  • Page 2, line 81: Please change “anti-inflammation effect” into “anti-inflammatory effect”.
  • Page 5, line 177: Please specify the type of DMEM. Did you use low or high glucose DMEM?
  • Figure 3: Do the authors have an explanation why the RNA level is not concentration-dependent blocked, whereas NO and iNOS protein are?
  • Figure 3: Add kDa to the detected Western Blot bands.
  • Please use µM instead of
  • Figure 5: Please use NF-κB instead of NκB.

Author Response

Thank you for your review and we sent answer for your review.

Reviewer 2 Report

Overall, this could be an interesting manuscript, as the authors showed that 5-hydroxymaltol derived from Lactobacillus fermentation may be an effective compound for treating inflammation-mediated diseases.  However, my main concern is that dose used are high and it is not clear why each assay have different time of treatment. Moreover, in most of the case there are not difference when the authors used 200, 500 or 1000 uM in the same assay. At this point this manuscript is not suitable to be considered for publication in Antioxidants.

  1. Authors did not explain why the selected all dose, and why in each assay they used different dose. Moreover, treatment time are different in M&M and figure legends. If the authors want to  apply this treatment in vivo experimental models and finally in human, what should be the dose used?
  2. Result section is very poor. Here the authors should include data to support quantitatively these results, and also include a general statement of significance, e.g., %xxx or xxx-fold
  3. In discussion section the authors should explain and discuss the different dose used and why there is not difference between 200, 500 and 1000 uM. 

Author Response

(The authors gave the same response as above.)

Reviewer 3 Report

The article of Kim et al. describes the isolation of a compound with antioxidant properties from Lactobacillus fermentation-processed beet juice. Analysis of the compound identified it as 5-hydroxymaltol. Further testing the commercially obtained 5-hydroxymaltol for anti-inflammatory and anti-oxidant activity using RAW264.7 murine macrophage cell line, the authors found it to inhibit inflammatory responses of RAW264.7 cells to bacterial lipopolysaccharide (LPS). This included inhibition of LPS-induced cytokine mRNA and protein expression, nitric oxide production, and NFkB and MAPK signaling. Moreover, 5-hydroxymaltol suppressed LPS-triggered reactive oxygen species formation and induced nuclear accumulation of Nrf2 transcription factor and its target heme oxygenase-1.

The study has no design flaws, and the experiments are clearly described.

However, the conclusion that “5-hydroxymaltol induced protein level of Nrf2 and HO-1 and elevated Nrf2/HO-1 signaling induce anti-inflammatory response to LPS treatment by reducing oxidative stress in macrophage” is not supported by the data, since Nrf2 activation is observed at earliest after 3 hours post-5-hydroxymaltol treatment whereas inhibition of LPS-induced MAPK and NFkB signaling occurs already after 1-hour treatment with 5-hydroxymaltol. Therefore, the ROS-related and anti-inflammatory effects of 5-hydroxymaltol are likely decoupled.

Also, throughout the manuscript, substantial language editing is needed.

Specific comments:

Figure 5B. Nrf2 p65 microscopy data are not convincing and should be completely removed, nuclear fractionation data (5A) are enough.

Abstract (lane 21) and Page 7, lane 259: “1% beet power” – should mean powder?

5-hydroxymaltol is written 5-hydoxymaltol or 5-hydromaltol on many occasions throughout the text

Page2, Lane 68 – should betalamic mean betalamic acid? Specify betalain – is it betanin?

Author Response

(The authors gave the same response as above.)

Reviewer 4 Report

The paper should be of interest but some minor and mjor points must be clarified;

Line 281: for 24h instead of “fore 1 day” would be more correct

Line 287; “western” should be “Western”

How the authors explain the difference in INOS inhibition between mRNA evaluation and protein expression?

The source of LPS, as well as serotype, should be indicated. The production of 40µM for 1µg/ml of LPS is quite high, the esperimental conditions should be better described (e.g. the number of cells/well)

Figure 5: IkB-α

Author Response

Thanks a lot for your good comments.

We sent answer for reviewer 4 comments.

Round 2

Reviewer 1 Report

All comments were adequately addressed by the authors.

Author Response

Thank you for your comments.

Reviewer 2 Report

Most of my main comments have not been addressed correctly. Still is not clear why the authors used that dose, why the dose used are very high and why each assay has different time of treatment. Moreover, there is not appropriate explication of why in most of the case there are not difference when the authors used 200, 500 or 1000 uM in the same assay.

Author Response

Thank you for your comments and we submitted answers of your commnets.

Reviewer 3 Report

The authors adequately addressed my comments and suggestions.

Author Response

Thank you for your comments

Reviewer 4 Report

The authors provided the requested changes

Round 3

Reviewer 2 Report

The authors have included some explication about why they used that dose, why the dose used are very high and why each assay has different time of treatment. However, I still think that there is not appropriate explication of why in most of the case there are not difference when the authors used 200, 500 or 1000 uM in the same assay.

Author Response

Thanks a lot for your good comments.

We sent answer for reviewer 2 comments.
